# The evolution of substrate discrimination in macrolide antibiotic resistance enzymes

Andrew C. Pawlowski[1], Peter J. Stogios[2,3], Kalinka Koteva[1], Tatiana Skarina[2,3], Elena Evdokimova[2,3], Alexei Savchenko[2,3,4] & Gerard D. Wright[1]

The production of antibiotics by microbes in the environment and their use in medicine and agriculture select for existing and emerging resistance. To address this inevitability, prudent development of antibiotic drugs requires careful consideration of resistance evolution. Here, we identify the molecular basis for expanded substrate specificity in MphI, a macrolide kinase (Mph) that does not confer resistance to erythromycin, in contrast to other known Mphs. Using a combination of phylogenetics, drug-resistance phenotypes, and in vitro enzyme assays, we find that MphI and MphK phosphorylate erythromycin poorly resulting in an antibiotic-sensitive phenotype. Using likelihood reconstruction of ancestral sequences and site-saturation combinatorial mutagenesis, supported by Mph crystal structures, we determine that two non-obvious mutations in combination expand the substrate range. This approach should be applicable for studying the functional evolution of any antibiotic resistance enzyme and for evaluating the evolvability of resistance enzymes to new generations of antibiotic scaffolds.

[1] Michael G. DeGroote Institute for Infectious Disease Research and the Department of Biochemistry and Biomedical Sciences, McMaster University, Hamilton, L8S 4L8 ON, Canada. [2] Department of Chemical Engineering and Applied Chemistry, University of Toronto, Toronto, M5S 3E5 ON, Canada. [3] Center for Structural Genomics of Infectious Diseases (CSGID), Calgary, T2N 4N1 AB, Canada. [4] Department of Microbiology, Immunology and Infectious Diseases, University of Calgary, Calgary, T2N 4N1 AB, Canada. Correspondence and requests for materials should be addressed to G.D.W. (email: wrightge@mcmaster.ca)

Antibiotic resistance is difficult to overcome because its evolution is persistent and multifactorial[1,2]. In addition to genetic mutations that confer drug insensitivity, mobile genetic elements disseminate resistance genes in pathogens, adding to the complexity and unpredictability of the emergence of resistance in bacterial populations[3–5]. Moreover, the prediction and detection of emerging resistance phenotypes resulting from mutations in resistance genes themselves remains a barrier to clinical surveillance and strategic antibiotic use[6–9], along with antibiotic drug development[10,11]. A focus of the antibiotic pharmaceutical industry is on identifying new antibiotic scaffolds with decreased susceptibility to current and widespread resistance mechanisms[12–15]. Another fruitful avenue in drug discovery is the development of successive generations of existing antibiotic scaffolds that bypass antibiotic-inactivating enzymes and other resistance mechanisms[13,14,16,17]. For these strategies to be productive in the face of increasing rates of resistance, the functional and evolutionary landscape of resistance enzymes must be interrogated to inform new drug discovery efforts.

Macrolides such as azithromycin are among the most successful and highly prescribed antibiotics in the world. They are first-line treatments for community-acquired respiratory tract infections and gonorrhea[18], and increasingly used to treat infections caused by multi-drug resistant Enterobacteriaceae[19]. Actinobacteria produce numerous variants of this antibiotic class but clinical implementation is mostly limited to erythromycin and its semi-synthetic derivatives, azithromycin and clarithromycin. The defining features of macrolide antibiotics include a 12-membered to 16-membered macrolactone ring and a dimethylamino sugar linked to the C5 position, which is essential for interacting with its target;[20–22] the large subunit of the bacterial ribosome (Fig. 1). Some macrolides are decorated with additional sugars at C3 (azithromycin, erythromycin), 4′-OH of the dimethylamino sugar (carbomycin, josamycin), or at multiple positions on the macrolactone (megalomicin, spiramycin, tylosin). Telithromycin is a semi-synthetic derivative of erythromycin and a first generation ketolide with a C11–C12 carbamate and an N-linked alkyl–aryl

substituent, and a ketone (*keto*lide) at C3 in place of a cladinose. The inspiration for removing the cladinose in telithromycin development was natural ketolides that bypass the induction of macrolide resistance[12]. Nevertheless, bacteria have evolved to overcome to this modification[23–25]. Macrolide resistance is abundant in pathogenic bacteria, and is most often the result of GTP-dependent macrolide kinases (Mph), ribosomal methyltransferases, or efflux pumps[26]. Mph enzymes inactivate macrolides by phosphorylating the 2′-OH of the essential dimethylamino sugar[27,28], preventing it from binding the ribosome, and providing the chemical rationale for the resistance phenotype.

The Eukaryotic-like kinase superfamily is a structural family related to Eukaryotic protein kinases, and includes antibiotic kinases that modify macrolides (Mph), aminoglycosides[29] (APH), and tuberactinomycins[30] (VPH, CPH). APHs have undergone substantial functional radiation, leading to homologs with different substrate-specificities and regio-specificities[31]. Four Mph homologs (MphA[32], MphB[33], MphC[34], MphE[35]) are mobilized in human pathogens and confer resistance to a wide range of macrolide substrates. In contrast, Mphs from non-pathogenic bacteria are genetically and functionally diverse. We recently described an Mph (MphI) from the environmental bacterium *Paenibacillus* sp. LC231 that does not confer resistance to macrolides with a C3 cladinose[36]. *Paenibacillus* sp. LC231 was isolated from Lechuguilla Cave where it was spatiotemporally separated from surface bacteria for over four million years. MphI shares high sequence identity (94%) to homologs found in related surface *Paenibacillus* sp., indicating the functional divergence of MphI is not recent. The *Bacillus cereus* group have two genetically and functionally distinct Mph enzymes; one that modifies a broad range of macrolides and another that cannot modify macrolides with 16-membered rings[37]. The closest experimentally validated homolog to MphI is MphJ (51% identity), which confers resistance to erythromycin and tylosin, but not to spiramycin or josamycin[38]. Mph enzymes are therefore adaptable resistance enzymes with complex evolutionary paths.

In this study, we show that narrow range Mphs are common in Bacillales, and their phosphorylation rates of cladinose containing macrolides is insufficient to confer resistance. We reconstruct the evolutionary path to Mph functional divergence using ancestral sequence reconstruction and identify four candidate residues responsible for substrate specificity. Applying phylogenetics, structural biology, protein engineering, and in vitro enzyme assays to the Mph family, we evolve increased substrate range in MphI, which results from multiple non-obvious mutations that, in tandem, increase the catalytic rate towards cladinose containing macrolides. This work demonstrates that while the antibiotic kinases have evolved several distinct functions, there exists significant barriers to acquiring a very related, but non-intrinsic function.

**Fig. 1** Structure of macrolide antibiotics. The blue arrow indicates the site of phosphorylation by Mph enzymes. The C3 position is highlighted with a blue line

## Results

**Multiple independent macrolide kinase mobilization events**. Of the 13 Mph homologs that have been experimentally validated for macrolide kinase activity[27,32–37,39–41], 7 are captured by mobile genetic elements. This implies that Mphs are at a high risk for mobilization (Supplementary Fig. 1, Supplementary Table 1). These Mphs share low sequence identity and it is therefore likely that each homolog was mobilized independently; if they originated from a single mobilized Mph, then they should cluster together in phylogenetic analysis. However, only MphE, MphF, and MphG are phylogenetically clustered, while MphA, MphB, MphC, and MphN are isolated from other known Mphs (Fig. 2; Supplementary Fig. 2). These results show that Mph genes may

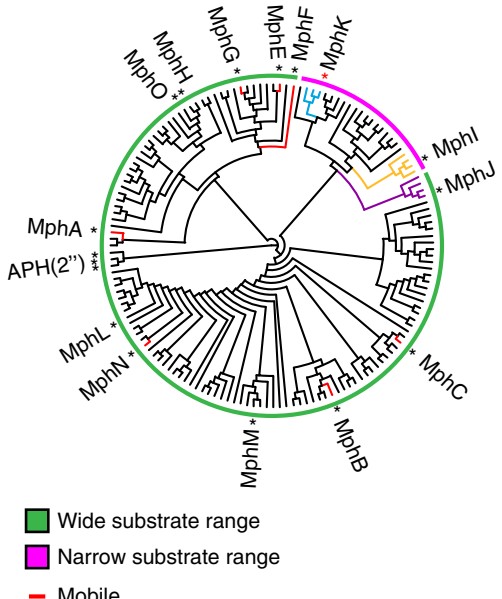

**Fig. 2** Phylogenetic reconstruction of macrolide phosphotransferases. Colored bars represent predicted substrate specificity based on phylogenetic clustering of known Mphs and their homologs. Asterisks represent experimentally validated Mphs, and the red asterisk represents MphK, which was identified from this analysis as a Mph that does not confer C3 cladinose macrolide resistance. Colored portions of the phylogenetic tree highlight clusters of interest. Mphs captured on mobile genetic elements are colored red. The tree is presented as a cladogram, and the root drawn with GTP-dependent APH(2″) enzymes as outgroups. A cladogram with complete labels and bootstrap values can be found in Supplementary Figure 2

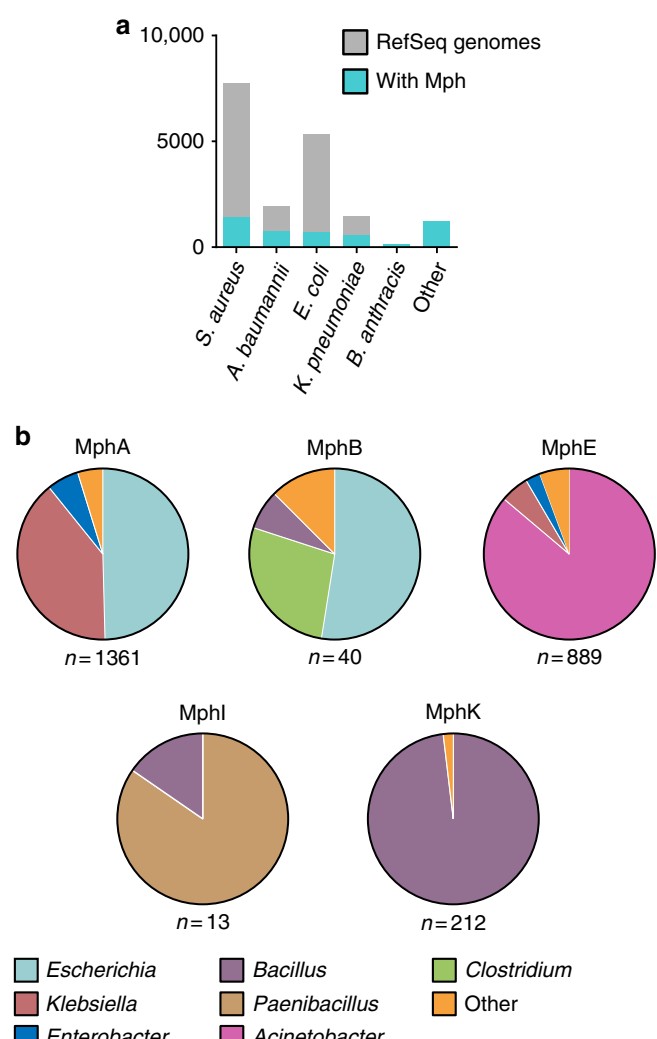

**Fig. 3** Taxonomic distribution of Mphs. **a** The top five bacterial species, in the RefSeq genome database, with Mphs by abundance, and presented in a stacked bar graph. **b** Taxonomic distribution of MphA, MphB, MphE, MphI, and MphK by genera

have been captured by mobile genetic elements at least five independent times.

The taxonomic distribution Mphs remains unclear despite several studies detailing their phenotypic, biochemical, and structural characterizations. We expanded the known genetic and taxonomic diversity of Mph enzymes by surveying bacterial strains with genomes deposited in the RefSeq database. Mphs were detected in 4732 genomes across 71 genera, and were predominantly found in *Staphylococcus, Acinetobacter, Escherichia, Bacillus*, and *Klebsiella* (Supplementary Fig. 3). Gram-negative pathogens, such as *A. baumannii* (38% of all *A. baumannii* strains), *K. pneumoniae* (38%), and *E. coli* (13%) are significantly enriched for Mphs (Fig. 3a). Ancestral *A. baumannii, K. pneumoniae*, and *E. coli* strains did not harbor an Mph homolog, and therefore it is very likely that these were acquired on mobile elements. Analysis of the taxonomic distribution of each Mph homolog reveals that MphA, MphB, and MphE are widespread in Gram-negative bacteria, and that the MphI homologs are limited to *Bacillus* and *Paenibacillus* and have not been mobilized (Fig. 3b; Supplementary Fig. 4). Our analysis demonstrates that Mphs are widely distributed across bacterial taxa, and the mobilized homologs are accumulating in Gram-negative pathogens.

**Substrate specificity correlates with genetic diversity.** MphI is the only known Mph that does not confer C3 cladinose macrolide resistance[36]. If MphI lost the ability to phosphorylate macrolides with a C3 cladinose, then we would expect it to share a common ancestor with Mphs that have wider substrate specificity, and therefore not positioned near the root of a phylogenetic reconstruction. The closest known homolog to MphI, MphJ (51%

sequence identity), can phosphorylate C3 cladinose macrolides[38]) and is an outgroup to the MphI clade (Fig. 2; Supplementary Fig. 2). The most parsimonious explanation is that MphI lost the ability to phosphorylate C3 cladinose macrolides. Adjacent to MphI is another clade with uncharacterized enzymes from *Bacillus* that we reasoned may show MphI-like specificity if phylogeny predicts substrate specificity. We heterologously expressed one of the members of this clade in *E. coli* TOP10 (*mphK/ycbJ*, 54% identical to *mphI*) from *Bacillus subtilis* 168 and verified that it confers resistance to macrolide antibiotics (Supplementary Table 2). Moreover, like *mphI, mphK* does not confer resistance to C3 cladinose macrolides, indicating that Mph substrate specificity correlates with phylogeny.

To further investigate Mph substrate specificity, we performed in vitro enzyme assays with purified MphB, MphI, and MphK. Azithromycin and telithromycin are semi-synthetic derivatives of erythromycin, with telithromycin lacking a C3 cladinose (Fig. 1). MphI and MphK both phosphorylate telithromycin but also phosphorylate azithromycin (Supplementary Fig. 5, Supplementary Table 3), suggesting that resistance phenotype may not correlate with biochemical analysis of drug modification. To confirm this result, we used tandem mass spectrometry to identify the site of phosphorylation on erythromycin, a macrolide

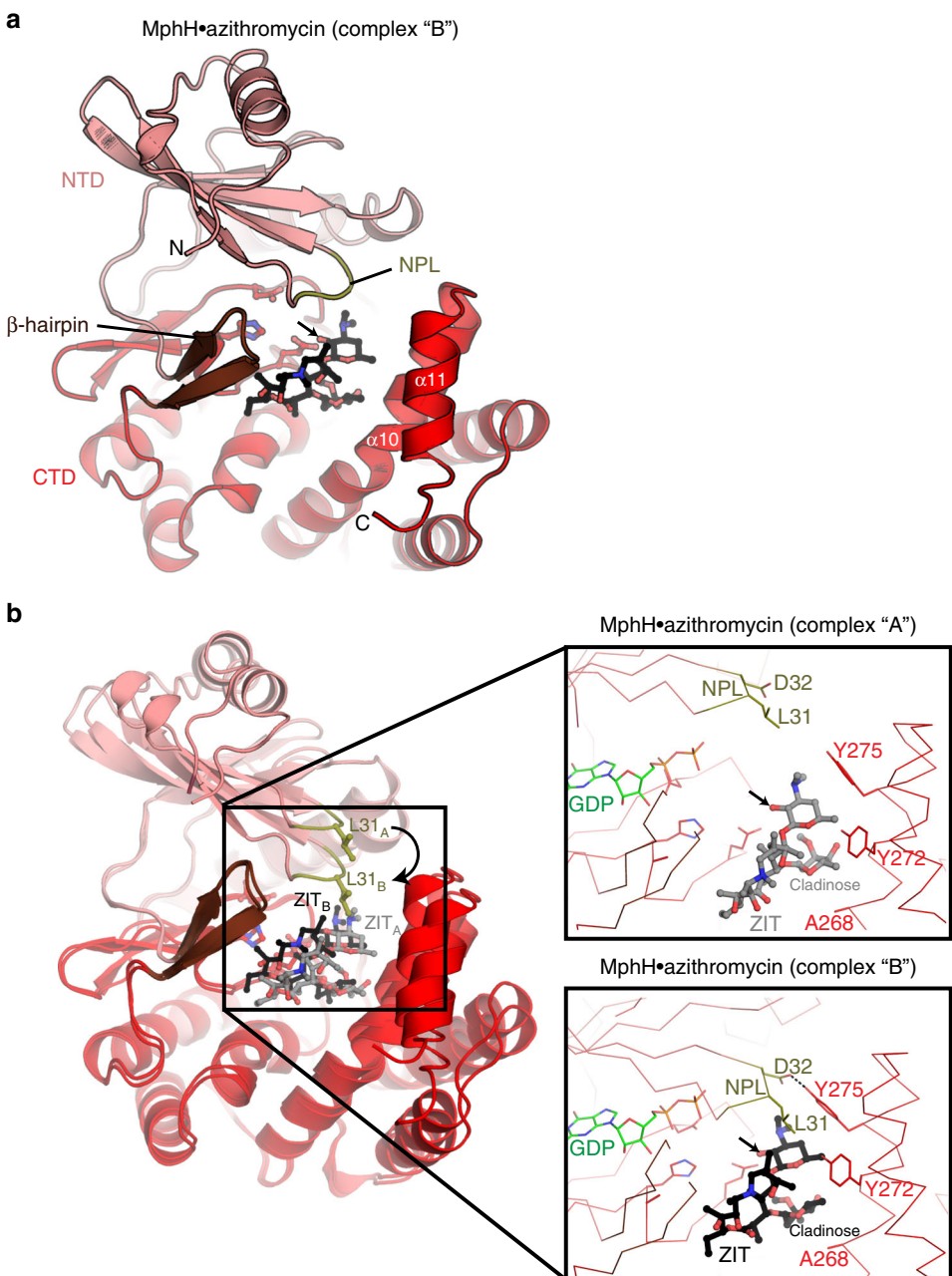

**Fig. 4** Crystal structures of MphH. **a** The MphH•azithromycin (complex "B" of the two complexes in the asymmetric unit of this crystal) is shown in cartoon, with NTD, CTD, NPL, and β-hairpin regions colored differently. Azithromycin (ZIT) is shown in black ball-and-stick representation. α10 and α11 which make up one face of the macrolide-binding site are labeled. Catalytic residues Glu196, His201, and Asp214 are shown in ball-and-stick. Arrow indicates 2'-OH site of phosphorylation on azithromycin. **b** Overlay of MphH•azithromycin complexes "A" and "B", showing conformational differences in NPL (particularly Leu31), position of azithromycin and α11. Inset shows zoom of macrolide-binding sites of each complex, showing key residues that contact the antibiotic compounds. Arrows indicate 2'-OH site of phosphorylation on azithromycin. GDP is shown from crystal structure of MphH•GDP complex

antibiotic with a similar structure to azithromycin and with well-established mass fragmentation patterns[42]. Both MphI and MphK phosphorylate the 2'-OH of erythromycin (Supplementary Fig. 6). Using steady-state kinetics, we demonstrated that MphI and MphK can indeed use C3 cladinose macrolides as substrates, but not efficiently ($k_{cat}/K_m$ $10^1$–$10^2$) (Supplementary Tables 4 and 5). The difference in catalytic constant ($k_{cat}/K_m$) values between C3 cladinose macrolides and descladinose macrolides is about $10^3$ and $10^1$–$10^3$ respectively for MphI and MphK. This poor biochemical specificity translates to the lack of resistance phenotype to these antibiotics when the genes are expressed in *E. coli* TOP10. To confirm whether this low activity against C3

cladinose macrolides impacts resistance phenotypes, we expressed *mphI* and *mphK* in an antibiotic hyper-susceptible strain of *E. coli* (*E. coli* BW25113 Δ*bamB*Δ*tolC*[43]). Both *mph* genes confer only a modest decrease in sensitivity to these antibiotics due to impaired efflux and outer membrane composition in this strain, verifying that the enzymatic activity observed in vitro has minimal impact on resistance phenotype (Supplementary Table 6).

MphB is another homolog also reported to have a narrow substrate range, modifying erythromycin but not azithromycin[44]. In contrast to previous reports, we found that MphB confers resistance to all macrolides tested (Supplementary Tables 2 and 3), and inactivates both telithromycin and azithromycin

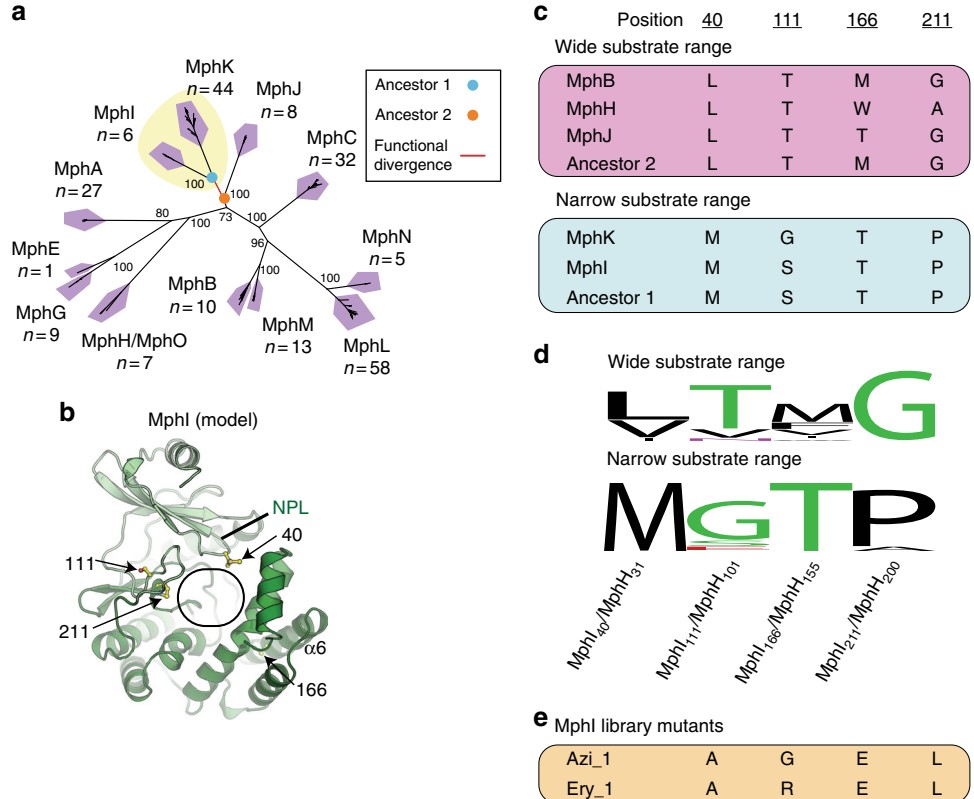

**Fig. 5** Ancestral sequence reconstruction of narrow-substrate and wide-substrate range Mphs for identifying possible determinants of substrate specificity. **a** Closely related homologs (>80% identity) to known Mphs were used to construct a phylogenetic tree. The tree is presented as an unrooted radial phylogram. The red tree branch represents the point of functional divergence between Mphs that can phosphorylate C3 cladinose macrolides, and those that cannot. The colored circles represent ancestral nodes with reconstructed sequences used to compare important residues. Bootstrap values are indicated at branch points. Numbers underneath Mph names represent the number of unique sequences in each cluster. The yellow background represents clades that do not efficiently phosphorylate macrolides with C3 cladinose. **b** Model of MphI based on crystal structure of MphH. Oval represents macrolide-binding pocket as determined from MphH●azithromycin crystal structures. Residues identified by ancestral sequence reconstruction of MphI–MphK divergence are colored in yellow. **c** Amino acid residues at select positions of narrow-spectrum and wide-spectrum Mphs, and their reconstructed ancestors. **d** Conservation of residues across the two Mph functional subtypes. Sequence conservation is displayed as a sequence logo using WebLogo. Wide substrate range includes sequences in all clades except MphI and MphK; narrow substrate range includes sequences in only MphI and MphK clades. **e** Amino acid residues at select positions of MphI library mutants with expanded substrate range

| Table 1 Antibiotic susceptibility of *mphI* and two library mutants with expanded resistance to C3 cladinose macrolides expressed in *E. coli* BW25113 Δ*bamB*Δ*tolC* | | | | |
|---|---|---|---|---|
| **Antibiotic** | **Empty vector** | *mphI* | *azi_1.1*[a] | *ery_1*[b,c] |
| Erythromycin | 0.5 | 1 | 16 | 16 |
| Clarithromycin | 0.5 | 1 | 4 | 4 |
| Azithromycin | 0.063 | 0.25 | 2–4 | 2–4 |
| Telithromycin | 0.13 | >16 | >16 | >16 |
| Spiramycin | 2 | 256 | 256 | 256 |
| Tylosin | 2 | 1024 | 1024 | 1024 |
| Josamycin | 2–4 | >64 | >64 | >64 |
| Kanamycin | 1 | 1 | 1 | 1 |

[a]MphI_M40A/S111R/T166E/P211L
[b]MphI_M40A/S111G/T166E/P211L
[c]MphI mutant Ery_1 was found to contain mutations in the promoter region, potentially altering expression and resistance phenotype. Ery_1 was therefore subcloned into the wild-type pGDP4 empty vector. MICs in this table are directly comparable.

comparable to other Mphs ($k_{cat}/K_m$ $10^4$–$10^6$), and the $K_m$ for C3 cladinose macrolides is 19–69× lower than MphI (Supplementary Table 8). In aggregate, these results show that C3 cladinose macrolides are good substrates for MphB and other Mphs including MphA[32,44] and MphH[27] (Supplementary Table 2), but are poor substrates for MphI and MphK.

**Structural analysis of macrolide recognition by Mph enzymes.** To better understand the molecular specificity of Mph enzymes against macrolides, we determined the 3-D structure of MphB as the apoenzyme, and three forms of MphH: (1) apoenzyme, (2) in complex with GDP, and (3) in complex with azithromycin (Supplementary Table 4 details of X-ray crystallographic statistics). While preparing this manuscript, crystal structures of MphA and MphB were reported in complex with various macrolides, which we have included in our analysis of substrate specificity[45]. We also attempted to crystallize MphI and MphK as apoenzymes, with several GTP analogs, and with spiramycin, tylosin, and telithromycin, but we were unsuccessful.

As expected, the structures of MphB (from our study) and MphH showed that the enzymes adopt the bi-lobe kinase fold reminiscent of APH(2″) enzyme crystal structures, with an N-terminal domain dominated by β sheets and a C-terminal domain

(Supplementary Table 6, Supplementary Fig. 5). Using tandem mass spectrometry, we also confirmed that MphB phosphorylates the desosamine 2′-OH of erythromycin (Supplementary Fig. 6). Moreover, MphB modifies macrolides with efficiencies

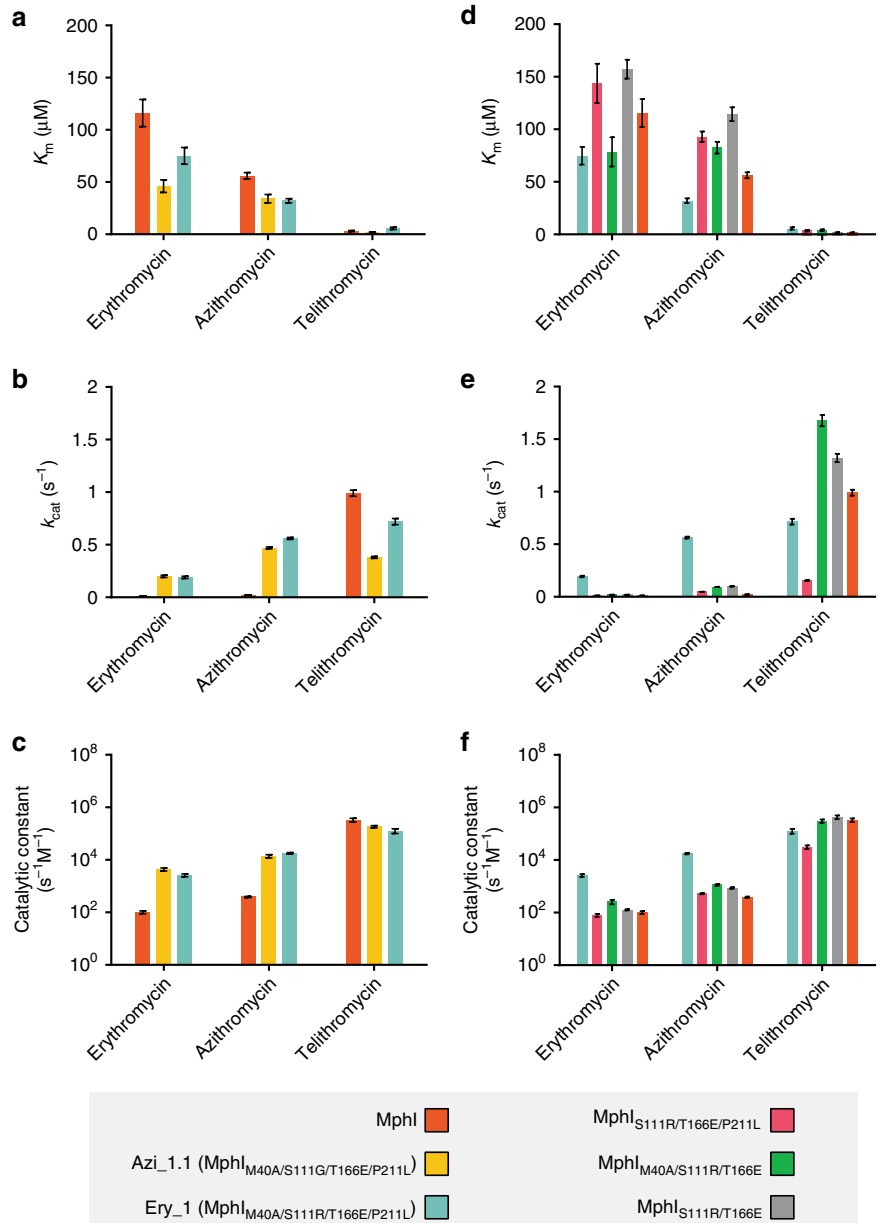

**Fig. 6** Kinetic characterization of MphI and mutants with improved activity towards C3 cladinose macrolides. Steady-state enzyme kinetics for MphI library mutants (**a**, **b**, **c**) and intermediate Ery_1 mutants (**d**, **e**, **f**). **a** $K_m$, **b** $k_{cat}$, and **c** $k_{cat}/K_m$ values for MphI and library mutants are presented in clustered bar graph format. **d** $K_m$, **e** $k_{cat}$, and **f** $k_{cat}/K_m$ values for intermediate Ery_1 mutants are also presented. Error bars represent the standard error of the mean (s.e. m.). The y-axis for **c** and **f** are presented in $\log_{10}$ format

comprised mainly of α-helices, linked via a β-hairpin insertion (residues 100–110) (Fig. 4a; Supplementary Fig. 7a).

The asymmetric unit of the MphH azithromycin crystal contains two copies of the complex ("complex A" and "complex B"), each showing subtle but important conformational differences (Fig. 4b). In both complexes, the azithromycin molecule is bound in a deep, electronegative cleft with one face formed by the end of C-terminal domain (i.e., helices α10 and α11), with the other face formed by the apposition of the β hairpin, the highly conserved region of the C-terminal domain containing the catalytic residues, and the nucleotide-positioning loop (NPL; residues 29–33) of the N-terminal domain (Fig. 4a). However, it was evident that the "complex B" encloses the bound macrolide more intimately and more appropriately for catalysis (Fig. 4b) and this complex dominates our analysis.

We compared the overall structure of MphH with MphA (5IGI[45]) and MphB (from ref. [45], 5IGV) to identify possible conformational changes associated with macrolide binding. While the MphA, MphB, and MphH structures are very similar, we observed structural distinctions in all three enzymes that localized to the (a) NPL, (b) β-hairpin insert, and (c) multiple residues in the macrolide-binding cleft (Supplementary Fig. 7a). As we observed in MphH, the NPL, and β-hairpin regions of both MphA and MphB adopt different conformations with macrolide binding. Consistent with the flexibility of MphH in our crystal structure, these observations highlight that conformational changes in these key regions of Mph enzymes play a role in macrolide substrate binding.

We compared the positioning of bound macrolides among the Mph enzymes to gain better insight into recognition of the C3 cladinose ring. The position of azithromycin in the macrolide-

**Table 2 Antibiotic susceptibility of intermediate mutants between MphI and MphI$_{M40A/S111R/T166E/P211L}$ expressed in *E. coli* BW25113 Δ*bamB*Δ*tolC***

| Antibiotic | ery_1 (MphI$_{M40A/S111R/T166E/P211L}$) | mphI$_{S111R/T166E/P211L}$ | mphI$_{M40A/T166E/P211L}$ | mphI$_{M40A/S111R/P211L}$ | mphI$_{M40A/S111R/T166E}$ | mphI$_{S111R/T166E}$ | mphI$_{M40A/P211L}$ | mphI[a] |
|---|---|---|---|---|---|---|---|---|
| Erythromycin | 32 | 4 | 32 | 32 | 4 | 2–4 | 32 | 2–4 |
| Clarithromycin | 8 | 1 | 8 | 8 | 2 | 1 | 16 | 1 |
| Azithromycin | 8 | 1 | 8 | 8 | 2 | 0.5 | 4–8 | 0.5 |
| Telithromycin | >16 | >16 | >16 | >16 | >16 | >16 | 32 | >16 |
| Spiramycin | 256 | 256 | 256 | 256 | 256 | 256 | 128–256 | >256 |
| Tylosin | 1024 | 1024 | 1024 | 1024 | 1024 | >1024 | >1024 | >1024 |
| Josamycin | >64 | >64 | >64 | >64 | >64 | >64 | >64 | >64 |
| Kanamycin | 1 | 1 | 1 | 1 | 1 | 1–2 | 1 | 1 |

[a]MphI mutant Ery_1 was found to contain mutations in the promoter region, potentially altering expression and resistance phenotype. MphI was subcloned into the Ery_1 vector and therefore MICs in this table are directly comparable.

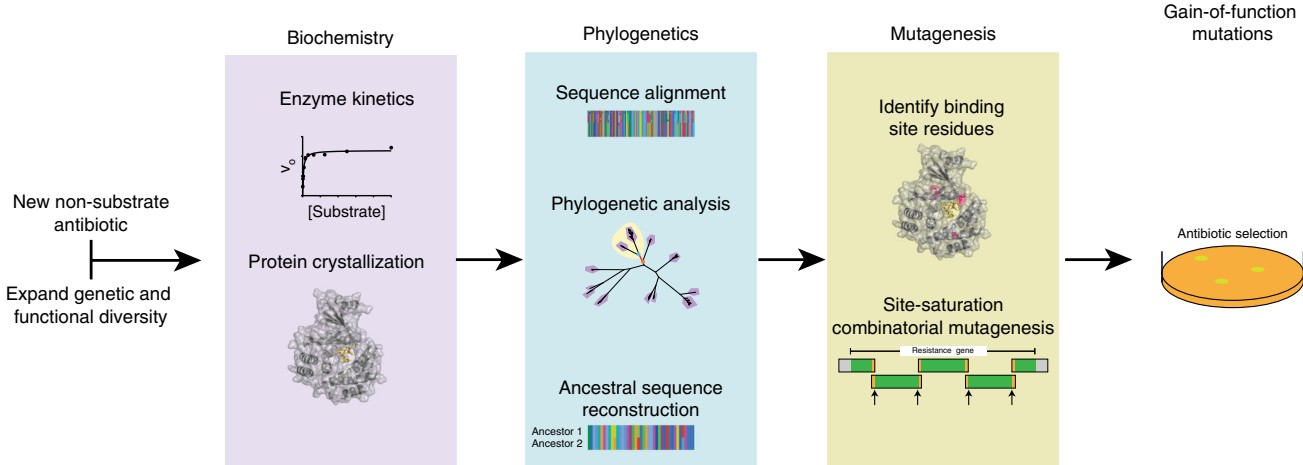

**Fig. 7** Workflow for assessing the evolvability of resistance enzymes to new antibiotics using biochemistry, phylogenetics, and mutagenesis. First, expand the known functional diversity of an enzyme class by identifying homologs that include the new antibiotic in its substrate range. Characterize the enzymes using steady-state kinetics and solve the protein structure of at least one homolog. Use the biochemical experiments to correlate functional diversity with phylogenetic diversity for identifying the last common ancestor of each functional subtype (e.g., ancestor of narrow range Mphs and wide range Mphs, respectively). The difference in ancestral protein sequences are residues possibly involved in substrate specificity, which should be prioritized based on proximity to the substrate-binding site in the protein structure. Target these residues for mutagenesis, preferably by saturation mutagenesis at each position, in combination (site-saturation combinatorial mutagenesis). Identify residues expanding substrate specificity by sequencing technologies that generate reads covering the whole gene (e.g., Sanger, Nanopore, PacBio)

binding pockets of MphA, MphB, and MphH are similar in that the 2' rings are oriented appropriately for catalysis, and the macrolactone ring interacts with the β-hairpin regions. However, the azithromycin molecule bound to MphH is positioned ~3 Å closer to the nucleotide binding region as compared to its position in either the MphA or MphB active sites (Supplementary Fig. 7a). Consequently, the C3 cladinose ring of azithromycin occupies deeper regions of the binding clefts of MphA or MphB. This analysis reveals that the C3 cladinose ring is positioned differently upon binding to individual Mph enzymes.

**Substrate selectivity is an emergent property in Mphs.** Since structural analysis demonstrated that the C3 cladinose occupies slightly different positions in the MphA, MphB, and MphH active sites, we speculated that MphI and MphK may have a large residue facing toward the C3 cladinose that would sterically hinder binding of these macrolides. We therefore compared the sequence composition of the six residues that approach the cladinose in the Mph structures (i.e., in the MphH azithromycin co-

structure—Glu196, Phe229, Phe265, Ala268, Gly271, Tyr272) across a multiple sequence alignment of all known Mphs and their close homologs (>80% identity). We found no significant correlations between residue type and antibiotic substrate specificity (Fig. 4b; Supplementary Fig. 7b, Supplementary Fig. 8). Ultimately, this analysis of the Mph azithromycin molecular contacts shows that the physiochemical properties of residues in proximity with the C3 cladinose ring are not conserved even in wide substrate range Mphs. Therefore, C3 cladinose macrolide specificity does not appear to be directed by nearby residues.

Since Mph substrate specificity is a phylogenetic trait, we reasoned that reconstructing the evolutionary path of functional divergence could identify the molecular determinants influencing C3 cladinose phosphorylation[46–50]. To this end, we reconstructed ancestral amino acid sequences[51] using extant enzymes based on a multiple sequence alignment and phylogeny of related Mph homologs (>80% identity) (Fig. 5a). Both Mph phylogenies (Figs. 2 and 5a) independently indicate MphI and MphK lost activity towards C3 cladinose macrolides, increasing our confidence in the branching order of Mph sequences. Therefore,

the equivalent residues different between Ancestor 1 (MphI and MphK) and Ancestor 2 (broad substrate range) may be determinants of substrate specificity[46]. We mapped these positions onto the MphH•azithromycin crystal structure and found four common sites near the macrolide-binding pocket, thus raising the possibility of their involvement in increased activity towards C3 cladinose macrolides (Fig. 5b–d). According to the MphH•azithromycin structure, one of these sites, MphH$_{L31}$ (MphI$_{M40}$) interacts with azithromycin; MphH$_{L31}$ faces toward the desosamine and C6–C7 of the macrolactone ring. MphH$_{A200}$ (MphI$_{P211}$) is on the short helix α8 that approaches MphH$_{Y198}$, which forms a key hydrophobic packing interaction with the macrolactone ring close to the methyl on C13 and the ethyl group on C14. MphH$_{W155}$ (MphI$_{T166}$) is in α6 that lines the macrolide-binding pocket and lies within 6 Å of the cladinose. The final site, MphH$_{T101}$ (MphI$_{S111}$) is more peripheral to the active site and is in the β hairpin but faces away from the macrolide-binding pocket.

We independently mutated each of these positions in MphI to the corresponding residue in the ancestral sequence; none of these variants conferred resistance to C3 cladinose macrolides (Supplementary Table 9). Since engineering new substrate specificity (gain of function) into enzymes is often complex and mutations may require a specific constellation of multiple residues to result in a functional change[52,53], we created an MphI library with mutations at all four positions, in combination, using combinatorial site-saturation mutagenesis. This enabled effective sampling all 160,000 possible amino acid substitutions at each of the four positions identified in ancestral reconstruction, in combination (1.05 million codon combinations). Using the powerful positive selection for antibiotic resistance to screen this mutant library, we were able to identify gain of function azithromycin or erythromycin resistance. We identified three azithromycin-resistant variants (Azi_1.1, Azi_1.2, Azi_1.3) with identical amino acid sequences, but different codon combinations, and one erythromycin-resistant variant (Ery_1) (Fig. 5e, Table 1; Supplementary Table 10). All variants with an expanded substrate range were identical at three residue positions: Ala40, Glu166, and Leu211 (mutants of MphI Met40, Thr166, Pro211). This result reveals a very limited number of routes to increase substrate range.

**MphI substrate selectivity is correlated with catalytic rate.** We purified Azi_1.1 and Ery_1, and verified azithromycin and telithromycin phosphorylation in an in vitro enzyme assay (Supplementary Fig. 5). Mutations in other antibiotic modifying enzymes are known to alter regiospecificity[54], and the cladinose hydroxyl of azithromycin is a possible phosphate acceptor. We unambiguously identified the desosamine 2′-OH as the phosphorylation site of Azi_1.1 and Ery_1 using tandem mass spectrometry (Supplementary Fig. 6). Using steady-state kinetics, we demonstrated that the $k_{cat}/K_m$ of Azi_1.1 and Ery_1 for C3 cladinose macrolides increased 35–86 fold and 26–46 fold, respectively (Fig. 6a–c; Supplementary Tables 11 and 12). Rather than a significant decrease in $K_m$ for C3 cladinose macrolides, the increase in activity towards these substrates was primarily driven by an increase in $k_{cat}$. The $K_m$ decreased by about half, while the $k_{cat}$ value increases 16×(erythromycin), 22–25×(azithromycin), and 48–51×(roxithromycin). Our results suggest that the four mutations in Azi_1.1 and Ery_1 increase the rate of product formation, and do not significantly increase catalytically productive C3 cladinose macrolide-binding affinity.

**Expanded substrate specificity requires binary mutations.** Ancestral sequence reconstruction successfully identified residues

implicated in substrate specificity, but the molecular roles of these residues remains unclear. To further explore these, we probed each position's contribution by systematically reversing each Ery_1 (MphI$_{M40A/S111R/T166E/P211L}$) mutation to wild-type sequence. MphI$_{M40A/T166E/P211L}$ and MphI$_{M40A/S111R/P211L}$ conferred resistance to C3 cladinose macrolides and retained the same resistance phenotype as Ery_1 (Table 2). Similarly, MphI$_{S111R/T166E}$ had the same resistance phenotype as MphI. These results demonstrate that positions 111 and 166 are not involved in expanding substrate specificity, consistent with their location more distal to the macrolide-binding site. In contrast, MphI$_{S111R/T166E/P211L}$ and MphI$_{M40A/S111R/T166E}$ lost the ability to confer C3 cladinose macrolide resistance, indicating that a binary combination of both mutations is required. MphI$_{M40A/P211L}$ had the same resistance phenotype as Ery_1 (MphI$_{M40A/S111R/T166E/P211L}$), further supporting the role of both mutations. We confirmed these results using steady-state kinetics of MphI$_{S111R/T166E/P211L}$, MphI$_{M40A/S111R/T166E}$, and MphI$_{S111R/T166E}$ (Fig. 6d–f; Supplementary Tables 13–15). Each MphI variant had $k_{cat}/K_m$ values similar to wild-type MphI. These results indicate that together, residues Met40 and Pro211, play key roles in macrolide substrate specificity of Mphs. Met40 and Pro211 are located in the nucleotide-positioning loop and in the highly structurally conserved region near GDP-coordinating/catalytic residues, respectively.

## Discussion

New generations of macrolide antibiotics are being developed to bypass resistance[17,55], which necessitates understanding the evolutionary landscape of Mphs. In this study, we find that two non-obvious mutations in tandem are required for expanding the resistance phenotype of MphI to include C3 cladinose macrolides by increasing the catalytic rate. These results are consistent with a plastic MphI active site that accommodates a wide selection of macrolides, but selectively phosphorylates macrolides without a cladinose. Narrow range Mphs such as MphI lost the ability to modify C3 cladinose macrolides, likely because of lack of selection by diverse macrolide structures in the local environment accompanied by neutral drift.

Amino acid residues positioned near, but not in, substrate-binding sites can impact function by modulating protein dynamics[56,57]. For example, combinations of mutations located outside the β-lactam binding pocket of β-lactamases expand substrate range by increasing conformational flexibility[50,56–58]. Introduction of each mutation individually does not expand the substrate range of β-lactamases[57]. In our study, M40A and P211L are located outside the MphI macrolide-binding pocket and both mutations are required to confer resistance to C3 cladinose macrolides. Altogether, the binary M40A and P211L mutations may increase the conformation flexibility of the macrolide-binding pocket, increasing the phosphorylation rate of C3 cladinose macrolides.

Understanding the evolution of resistance enzymes is of considerable interest because of their wide functional and genetic diversity, and their biological importance in medicine and microbial ecology[2,59,60]. Combining random mutagenesis strategies with antibiotic selection is powerful for interrogating the genetic-phenotypic continuum of resistance enzymes. Indeed, many of the strategies for protein engineering and synthetic biology were first developed using resistance enzymes[61–65] (e.g., TEM β-lactamase), and provide a strong foundation for biochemical investigations of resistance. Our approach builds on previous strategies by identifying target positions using phylogenetic and evolutionary principals, and sampling random genetic diversity at each of the four positions simultaneously. The

strategy we report here offers a single step for efficient site-saturation combinatorial mutagenesis of multiple positions using Gibson Assembly and productive sampling of the mutant library (>1 million variants) using positive antibiotic selection. Our approach successfully identified candidate positions involved in Mph substrate selectivity that were unidentifiable from Mph•macrolide co-structures reported here or by Fong et al.[45]. Despite the macrolactone ring and dimethylamino sugar shared by all macrolides, the M40A and P211L mutations in MphI specifically increase phosphorylation rate of those with a C3 cladinose without affecting catalysis of descladinose macrolides. In the MphH•azithromycin co-structure, the equivalent MphH positions Leu31 and Ala200 do not interact with the C3 cladinose. Our study illuminates the utility of evolutionary guided mutagenesis for studying non-obvious molecular determinants of substrate selectivity.

Ancestral sequence reconstruction has proven to be a valuable tool in drug discovery[49], protein engineering[46,48], and molecular evolution studies[47,50,66]. We believe our approach will be widely applicable for uncovering the molecular basis of functional divergence in antibiotic resistance enzymes. New generations of antibiotic scaffolds are being designed to bypass resistance mechanisms mobilized in pathogens[67]. However, if a homolog in environmental bacteria confers resistance to the new antibiotics, the potential for evolving resistance is high. Use of ancestral sequence reconstruction combined with structural and biochemical studies, as performed here, will be useful for predicting the possibility and feasibility of expanded resistance phenotypes to semi-synthetic antibiotics (Fig. 7).

Recent platforms that synthesize diverse macrolides will enable the discovery of newer generations of resistance-proof antibiotics[17,55]. Our study suggest that macrolide binding may be a precursor to phenotypic resistance in Mphs. Therefore, next generation macrolide antibiotics that bypass Mph-mediated resistance should also be investigated for active site binding, and whether homologs from environmental bacteria confer resistance.

## Methods

**Antibiotics and reagents**. All buffers and salts were purchased from BioShop (Burlington, ON, Canada) unless otherwise specified. Macrolide antibiotics, GTP, pyruvate kinase: lactate dehydrogenase (PK:LDH), phosphoenolpyruvate, and NADH were purchased from Sigma-Aldrich (Oakville, ON, Canada). Molecular biology kits and organic solvents were purchased from Fisher Scientific (Ottawa, ON, Canada). Telithromycin was purified from the drug formulation Ketek (400 mg, Sanofi-Aventis US)[36].

**Phylogenetic reconstruction of macrolide kinases**. Diverse Mph sequences were identified with BLASTp[68] by querying the Mph sequences in Supplementary Table 1 with an e-value cutoff of 1e−30, and the following flags; -max_target_seqs 500, and −entrez_query 'NOT partial'. Note that Mph enzymes were renamed for consistency with established nomenclature[26] and to avoid multiple distinct sequences (<80% amino acid identity) with the same name. A hit was defined as having at least 50% sequence identity and an alignment length >80% of the query length. Sequences were clustered at 80% identity using the cluster_smallmem algorithm of uclust[69], and aligned with MAFFT (L-INS-i method)[70]. APH(2″) enzymes are homologs of Mphs[71], and therefore APH(2″)-IIa, APH(2″)-IIIa, and APH(2″)-Ie were used as outgroups for rooting the tree. The alignment was then weighted by alignment confidence using the transitive consistency score (TCS) function of T-coffee[72]. RAxML was used to generate a maximum likelihood phylogenetic tree with the GAMMA model of rate heterogeneity and LG empirical base frequencies (-m PROTGAMMAAUTO) and using 100 rapid bootstraps[73].

**Mph cloning and antibiotic susceptibility testing**. mphB was codon optimized, synthetized as a gBlock (IDT) and cloned into pET28a with an N-terminal histidine tag. The codon optimized mphB sequence is in Supplementary Table 17. mphI was previously cloned into pET22b[36]. pGDP4 is a derivative of pET28a with the multiple-cloning site under the control of the constitutive lac promoter[43]. mphI was subcloned into pGDP4 using the XbaI and XhoI restriction sites. E. coli was routinely cultured on LB-Lennox (Bioshop) at 37 °C overnight, and either 100 µg mL[−1] ampicillin or 50 µg mL[−1] kanamycin was used for plasmid selection.

E. coli TOP10 (Invitrogen) was used for routine cloning experiments and susceptibility testing of mphs cloned into pGDP4. E. coli BL21(DE3) (Stratagene, USA) was used for protein overexpression experiments. E. coli BW25113 ΔbamBΔtolC is an antibiotic hypersensitive[43], and was used for susceptibility testing of Mphs cloning into pGDP4. Antibiotic susceptibility testing was performed according to the Clinical and Laboratory Standards Institute guidelines[74] for determining minimal inhibitory concentration (MIC) in 96-well plates (Sarstedt, Germany) in duplicate. MIC values were reproducible. A saline suspension of OD600 of 0.1 was made from 2–3 colonies of a fresh overnight, and diluted 1:200 into Mueller-Hinton Broth (BD Biosciences). MIC values were read after stationary incubation at 37 °C for 24 h. Susceptibility assays were performed at least two independent times.

**Protein overexpression and purification**. Mphs were purified as described previously for MphI[36]. E. coli BL21(DE3) was cultured with shaking at 37 °C in 1 L of LB-Lennox until an OD600 of 0.5–0.6, chilled in an ice bath for 20 min, and protein overexpression induced by the addition of 1 mM IPTG and incubation at 17 °C for 16 h. Cell mass was harvested by centrifuging at 10,000 × g for 20 min, washed with cold saline, and resuspended in 20 mL buffer A (50 mM HEPES pH 7.5, 150 mM NaCl, 5% glycerol, 10 mM imidazole). Cell lysis was performed using a One-shot Cell Disruptor (Constant Systems Limited) at 20,000 psi, and cell debris removed after adding 15 mL buffer A, 5 mg bovine bovine pancreas DNase, and 2.5 mg of bovine pancreas RNase by centrifugation at 50,000 × g for 45 min. Proteins were purified using 1 mL Ni[2+]-nitrilotriacetic acid column (Qiagen) and a linear gradient of 95% buffer A to 100% buffer B (buffer A with 250 mM imidazole) over 20 column volumes. Fractions with pure Mph were pooled and desalted into 50 mM HEPES pH 7.5 using a PD-10 desalting column (GE Scientific). Pure enzyme stocks were stored at 4 °C.

**Characterization of phosphorylated macrolides**. The product of MphB, MphI, MphK, Azi_1.1, and Ery_1 catalyzed reactions were confirmed with LC-MS analysis. Each 100 µL reaction consisted of Mph Buffer (50 mM HEPES pH 7.5, 40 mM KCl, 10 mM MgCl2), 1 mg mL[−1] azithromycin or telithromycin, 2 mM GTP, and 125–250 µg of enzyme. Control reactions were prepared identically, but devoid of enzyme. Reactions were incubated at 37 °C overnight, stopped with the addition of 100 µL cold methanol and stored at −20 °C. Before analysis, samples were centrifuged at 16,000 × g for 10 min. 20 µL of each reaction was injected onto an Agilent 1100 Series LC system and a QTRAP LC/MS/MS System (ABSciex) using the following HPLC conditions: isocratic 5% solvent B (0.05% formic acid in acetonitrile), 95% solvent A (0.05% formic acid in water) over 1 min, followed by a linear gradient to 97% B over 7 min at a flow rate of 1 mL min[−1] and C18 column (Sunfire, 5 µm, 4.6×50 mm).

**Structural elucidation of phosphorylated erythromycin**. Tandem mass spectrometry was used to identify the site of phosphorylation by MphB, Azi_1.1, and Ery_1. Each 100 µL reaction consisted of Mph Buffer, 0.5 mg mL[−1] erythromycin, 2 mM GTP, and 125–250 µg enzyme. Reactions were performed as described above. After centrifugation, samples were diluted 1:1000 into 50% methanol. High resolution electrospray ionization mass spectra were acquired using Agilent 1290 UPLC separation module and qTOF G6550A mass detector in positive ion mode. Fragmentor voltage was set to 365.0 V. Liquid chromatography separation was carried out using Eclipse C18 (3.5 µm, 2.1 × 100 mm) column (Agilent Technologies) and the following pump method: at 0 min 95% solvent A (0.1% v/v formic acid in water), from 1 to 7 min up to 97% solvent B (0.1% v/v of formic acid in acetonitrile), at a flow rate 0.4 mL min[−1]. Targeted MS[2] acquisition mode was used for the fragmentation under 3 different fixed collision induced dissociation (CID) settings: 10, 30 and 40 eV. Targeted list contained the following m/z: 734.4681 (erythromycin [M+H][+]) and 814.4342 (Mph erythromycin product [M+H][+]). Fragments were identified based on ref. [42].

**Steady-state kinetics of Mphs**. Steady-state kinetics for Mphs were measured using the continuous PK/LDH-coupled assay[27,32,36,71]. Reactions were performed in 96-well plates (Nunc, Thermo Scientific) using Spectramax Plus384 (Molecular Devices) in duplicate. Individual reactions were 100 µL and contained 50 mM HEPES pH 7.5, 40 mM KCl, 10 mM MgCl2, 0.2 mM NADH, 3.5 mM PEP, and 4.8 U PK/LDH (pyruvate kinase/lactate dehydrogenase). For determining the $K_m$ for GTP, telithromycin was maintained at 100 µM and GTP was varied between 0.49 and 250 µM. For determining the $K_m$ for macrolides, GTP was maintained at 200 µM and macrolides were varied between 0.49 µM and 1000 µM. Enzyme efficiency varied considerably between substrates, and therefore some enzymes were used at different concentrations depending on the substrate in order to produce a linear range suitable for $K_m$ determination. These concentrations were as follows; MphB 140 nM (most substrates), MphB 36 nM (josamycin), MphI 45 nM (descladinose macrolides), MphI 1.44 µM (C3 cladinose macrolides), Azi_1.1 90 nM (all substrates), Ery_1 (all substrates), MphK 680 nM (all substrates), MphI$_{S111R/T166E/P211L}$ 90 nM (descladinose macrolides), MphI$_{S111R/T166E/P211L}$ 1.43 µM (C3 cladinose macrolides), MphI$_{M40A/S111R/T166E}$ 22 nM (descladinose macrolides), MphI$_{M40A/S111R/T166E}$ 430 nM (C3 cladinose macrolides), MphI$_{S111R/T166E}$ 22 nM (descladinose

macrolides), $MphI_{S111R/T166E}$ 1.43 μM (C3 cladinose macrolides). Enzyme kinetics experiments were performed at least two independent times.

**Ancestral reconstruction of ancestral Mph sequences.** Close Mph homologs were identified with ncbi-blast-2.2.31 + BLASTp[68] by querying the Mph sequences in Supplementary Table 1, e-value cutoff of 1e-50, -max_target_seqs 500, and −entrez_query 'NOT partial'. A close homolog was defined as having >80% identity and an alignment length >85% of the query length. Sequences were aligned using MAFFT (L-INS-i) and the BLOSUM45 substitution matrix. The alignment was weighted using the T-coffee TCS function. A maximum likelihood tree was generated with RAxML with the JTT model (PROTGAMAAAUTO) and rapid bootstrap analysis of 100 replicates. Reconstructed ancestral sequences were inferred using an Empirical Bayes approach (PAML 4.9a) with the jones amino acid replacement matrix. For ancestral reconstruction, the RAxML generated tree was rooted at the point of functional divergence and used the unweighted multiple sequence alignment. The ancestral sequences of each Mph subtype (i.e., those that phosphorylate C3 cladinose macrolide efficiently and those that do it poorly) were contrasted, and these positions were mapped onto the MphH crystal structure to identify residues near the macrolide-binding pocket. The alignment for ancestral reconstruction was also used to identify amino acid conservation for sequence logos.

**Construction of MphI mutant library.** We constructed the MphI mutant library by simultaneously introducing 'NNK' codons at amino acid positions 40, 111, 166, and 211 using a hierarchical approach based on the Gibson Assembly method[75]. This method assembled 4 PCR fragments; 3 sections of *mphI* and another containing part of the gene and the vector. Oligonucleotides were designed to amplify 93–358 ($f_1$) and 471–633 ($f_3$), and incorporate 'NNK' codons at the target positions with an additional 25 bp on each end for Gibson Assembly. Additionally, 334–495 ($f_3$) was amplified, as well as the vector, 1–117, and 634–927 ($f_4$) (Supplementary Fig. 9). PCR fragments were amplified with Phusion Polymerase (Thermo Scientific) and gel purified with 2% low melting point agarose (Thermo Scientific). Fragments $f_1$ (266 bp), $f_2$ (162 bp), $f_3$ (188 bp), and $f_4$ backbone (3636 bp) were assembled in a 20 μL Gibson Assembly reaction (performed in triplicate) and incubated at 50 °C for 1 h. The Gibson reaction contained; 100 ng $f_4$ backbone, 20 ng $f_1$, 12 ng $f_2$, 14 ng $f_3$, 0.004 U T5 exonuclease (New England Biolabs), 0.025 U Phusion polymerase, 1 U Taq ligase (New England Biolabs), 0.2 mM each dNTP (FroggaBio), 1 mM NAD (Sigma), 100 mM Tris-HCl pH 7.5 (Bioshop), 10 mM $MgCl_2$ (Fisher), 10 mM DTT (Bioshop), and 125 mg mL$^{-1}$ PEG-8000 (Sigma). Reactions were pooled, purified using PCR purification kit (Thermo Scientific) and eluted in 20 μL water. A volume of 1 μL was used to transform three individual *E. coli* Electromax electrocompetent cell (Invitrogen) reactions. Cells were recovered in a total of 3 mL Super Optimal broth with Catabolite repression (SOC) (Invitrogen) for 1 h at 37 °C with shaking. Library titer was estimated by plating 10$^{-3}$ and 10$^{-4}$ dilutions and determining the CFU mL$^{-1}$ of transformation. Plasmids from 10 random colonies were isolated, sent for Sanger sequencing (Mobix, McMaster University), and verified to contain a unique missense mutation at each of the four target positions. In total, the library contained 3.9 million variants. The rest of the transformations were used to inoculate 100 mL LB-Lennox with 100 μg mL$^{-1}$ carbenicillin and incubated with shaking for 16 h at 37 °C. 50 mL was centrifuged at 4000 × *g* for 1 h at 4 °C, resuspended in 40 mL of fresh LB-Lennox and 10 mL of 80% glycerol, and stored at −80 °C. The library was diluted to OD$_{600}$ of 0.25 (~200 million CFU mL$^{-1}$) and 250 μL was plated (~51 × the library) on LB agar with 20 μg mL$^{-1}$ azithromycin and 100 μg mL$^{-1}$ carbenicillin. In another experiment, the library was diluted to OD$_{600}$ of 0.022 (~17.8 million CFU mL$^{-1}$) and 250 μL was plated (~4.6 × the library) on 250 μg mL$^{-1}$ erythromycin and 75 μg mL$^{-1}$ carbenicillin. The concentration of azithromycin and erythromycin was such that *E. coli* Electromax harboring wild-type *mphI* could not grow. Plasmids from azithromycin or erythromycin-resistant mutants were isolated, transformed into *E. coli* TOP10, and confirmed to be resistant to C3 cladinose macrolides. This verified that the resistance phenotype was encoded on the plasmid and not a chromosomal mutation. Missense mutations were identified by Sanger sequencing (Mobix, McMaster University). For protein purification experiments MphI mutants were subcloned into pET22b.

**Site-directed mutagenesis.** Oligonucleotides in Supplementary Table 16 were used for site-directed mutagenesis in the following 50 μL reaction; 10 ng plasmid, 0.5 μM primers, 200 μM dNTPs, 1 × HF buffer, 1 U Phusion polymerase. The following thermocycler conditions were used; initial denaturation 98 °C 2 min, 98 °C 10 s, 55 °C 20 s, 72 °C 3.75 min for 16 cycles, and 7 min final extension at 72 °C. An aliquot of 2 μL of FastDigest DpnI (Thermo Scientific) was added and incubated at 37 °C for 1 h to remove template DNA. Reactions were cleaned up with PCR purification, and 5 μL was used to transform chemically competent *E. coli* TOP10. Plasmids were isolated and sent for Sanger sequencing to confirm the desired mutations. Wild-type MphI was used for individual mutations predicted with ancestral reconstruction. The Ery_1 MphI mutant was used to reverse mutations to wildtype. All mutations were made independently in pET28a-*mphI*, pET28a-*ery_1*, pGDP4-*mphI*, and pGDP4-*ery_1*.

**Taxonomic distribution of Mphs.** The RefSeq genome database was used to evaluate the taxonomic distribution of Mphs in bacteria. The assembly report for all 82,640 RefSeq genomes (as of 29 March 2017) was used to correlate nucleotide accession numbers with RefSeq genomes. The identical protein report was acquired for each protein accession from the phylogenetic reconstruction of diverse Mphs using the efetch Entrez function. Strains with a Mph were identified if the nucleotide accession of a Mph was within the list of RefSeq genomes above. Strains with 2 Mphs were normalized to 1 by matching the unique 4 letter prefix of each nucleotide accession number for WGS sequences, or by matching the entire accession number for complete genomes. This was performed using custom python code.

**Crystal structure determination of Mphs.** For crystallization, Mph genes were subcloned into the pMCSG53 expression vector that codes for a N-terminal His$_6$-tag and a TEV protease cleavage site. Se-Met-substituted versions of MphB and MphH were expressed using the standard M9 high-yield growth procedure according to the manufacturer's instructions (Shanghai Medicilon). Crystallization was performed at room temperature using the sitting drop method and 2 μL protein or protein:ligand mixture plus 2 μL reservoir solution. The MphH concentration was 10 mg mL$^{-1}$ and the MphB concentration was 90 mg mL$^{-1}$. Se-Met MphB (apoenzyme) was crystallized with reservoir solution 0.2 M calcium acetate, 0.1 M sodium cacodylate pH 6.5, 18% (w/v) PEG 8 K. Native MphH (apoenzyme) was crystallized with reservoir solution 2 M ammonium sulfate, 0.1 M Hepes pH 7.5, 2% (w/v) PEG 400 and 1% trehalose. Se-Met MphH•GDP complex was crystallized with reservoir solution 2 M ammonium sulfate, 0.1 M Hepes pH 7.5, 2% (w/v) PEG400, 1% trehalose, 5 mM azithromycin and 5 mM GMPPCP. Native MphH•azithromycin complex was crystallized with reservoir solution 2 M ammonium sulfate, 5 mM azithromycin and 5% DMSO. All crystals were cryoprotected in paratone oil prior to data collection. Diffraction data for MphH (native, apoenzyme) was collected at 100 K using a Rigaku Micromax 007-HF rotating anode and a Rigaku R-AXIS IV ++detector. Diffraction data for MphB (Se-Met, apoenzyme), MphH•GDP (Se-Met) and MphH•azithromycin (native) were collected at the Advanced Photon Source, Argonne National Laboratory, Life Sciences Collaborative Access Team beamlines 21-ID-F or 21-ID-G at the selenium absorption edge. Data was processed by HKL-3000[76] or XDS[77] and Aimless[78]. SAD structure determination was completed using Phenix.autosol[79] and Molecular Replacement for native structures was compiled using Phenix.phaser. Refinement was completed with Phenix.refine and Coot[80]. All B-factors were refined as isotropic with TLS parameterization. All geometry was verified using Phenix validation tools and the wwPDB server. Electron density maps are in Supplementary Figure 10.

**Code availability.** The custom python code used to survey taxonomic distribution of Mphs is available from the corresponding author upon request.

**Data availability.** The crystal structures of Mph enzymes were deposited into the PDB; MphB Se-Met apoenzyme (5UXA), MphB native apoenzyme (5UXB), MphH (native) GDP + Pi complex (5UXC), and MphH (Se-Met) azithromycin complex (5UXD). Protein accession numbers for all Mph enzymes included in this study are listen in Supplementary Table 1. Other data are available from the corresponding author upon reasonable request.

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

## Acknowledgments

We thank Nicholas Waglechner (McMaster) for useful discussions on phylogenetics, Mph surveillance, and ancestral sequence reconstruction. We also thank Murray Junop (Western University) for helpful discussion on protein engineering experiments. This research was funded by a Canadian Institutes of Health Research grant (FRN-148463) and by a Canada Research Chair in Antibiotic Biochemistry (to G.D.W.) and through funds from the National Institute of Allergy and Infectious Diseases, National Institutes of Health, Department of Health and Human Services, under Contract No. HHSN272201200026C.

## Author contributions

A.C.P. and G.D.W. designed research. A.C.P., P.J.S., and K.K. performed research and analyzed data. A.C.P. performed the following experiments; antibiotic susceptibility testing, phylogenetics, enzyme kinetics, ancestral sequence reconstruction, mass spectrometry, site-saturation combinatorial mutagenesis, and in vitro enzyme assays. P.G.S. performed crystal structure determination of MphH. K.K. performed tandem mass spectrometry experiments. T.S. and E.E. performed protein crystallization. A.S. supervised protein crystallization and contributed to interpreting mechanism of expanded substrate range. A.C.P., P.J.S, and G.D.W wrote the manuscript with input from all authors.

## Additional information

**Competing interests:** The authors declare no competing financial interests.

