## [Peer Review File · Nature Communications]

Reviewers' comments:

Reviewer #1 (Remarks to the Author):

In this manuscript by Pawlowski et al. the authors make use of ancestral sequence reconstruction combined with structural and biochemical studies to determine how enzymes that modify macrolides evolve towards a broader substrate range. This is a well-written, clear and interesting manuscript that provides a strategy for evaluating the evolvability of resistance enzymes to new generations of antibiotic scaffolds that is of both practical (drug development) and more basic interest (evolution and mechanistic basis for enzyme substrate specificity).

Comments:

1. Page 5, the use of the word of sub-functionalization I do not think is proper here. Sub-functionalization in the context of evolution of new functions means that one function is partitioned between different gene copies (i.e. an interallelic complementation).

2. Page 7: "The most parsimonious explanation is that MphI lost the ability to phosphorylate C3 cladinose macrolides." Page 14: "Narrow range Mphs such as MphI lost the ability to modify C3 cladinose macrolides, likely because of lack of selection by diverse macrolide structures in the local environment accompanied by neutral drift."

Could another explanation not be that there was stronger selection for ability to provide resistance to specific macrolides and that this resulted in loss of activity towards some macrolides because of functional trade-offs?

3. Page 7, 8, "specificity constant" I would not call it a specificity constant. Rather k_{cat}/K_m is an efficiency factor but the ratio of the k_{cat}/K_m for substrate 1 and k_{cat}/K_m for substrate 2 provides a specificity measure.

4. Page 8 and forward. Was the structural analysis based on the authors data or the published data or both?

5. Page 10-11. Could shorten considerably the first paragraph on page 10-11 since the result is negative (i.e. the authors state that: "Therefore, C3 cladinose macrolide specificity does not appear to be directed by nearby residues."). Instead, the subsequent phylogenetic analysis provides an answer and one could go directly into that result.

6. Page 13. What is the structural/functional interpretation of the fact that the mutations mainly affected k_{cat} rather than K_m ?

7. Page 6, change reveal to reveals

8. Page 8 "...we attempted determined...." Should be either attempted or determined.

Reviewer #2 (Remarks to the Author):

Resistance to macrolide antibiotics is growing, threatening the usefulness of these essential antibacterial agents. Macrolide kinases (Mph) inactivate macrolides by phosphorylating an OH of the sugar moiety, preventing binding to the ribosome. Mph enzymes from non pathogenic bacteria are functionally diverse, being narrow range, in contrast to Mph's from pathogens. In this work, Pawlowski

et al. report the molecular basis for expanded substrate specificity in MphI.

A bioinformatic analysis of the distribution of Mphs reveals that Gram negative pathogens have acquired Mph genes through mobile elements. MphI is the only known Mph unable to confer C3-cladinose resistance. The authors study a series of Mph enzymes from *B. subtilis* and interrogate their substrate profile. MphI and MphK show poor (but not null) in vitro specificities to phosphorylate cladinose macrolides, that results in a small impact on the resistance phenotype. In contrast, MphB is broad spectrum and shows high catalytic efficiencies in modifying several macrolides.

The crystal structure of unbound MphB and ligated to azithromycin and GDP does not reveals clues about the specificity in active site residues. Ancestral sequence reconstruction provided clues to identify 4 positions that were randomized in MphI, resulting in the selection of mutants showing a clear gain of function in vitro. This has led to the identification of possible two mutational hotspots that may drive the expansion of substrate spectrum of Mphs.

This is an excellent piece of work, combining different techniques. There are, however, a few aspects that are unclear at this point:

1. The authors do not report the crystal structure of MphI, that would have been the ideal complement to the herein reported structures of MphB and MphH. Have they been unable to obtain crystals of MphI? Homology modelling may be insufficient.

2. The mutated residues are nearby, but not within the active site. Based on the crystal structures, the authors indirectly relate the impact of the substrate spectrum to conformational changes linked to substrate binding. Broadening of the substrate spectrum by second sphere mutations have been addressed by NMR (González et al (2016) *Mol Biol Evol.*;33, 1768-76) and Molecular Dynamics in beta-lactamases (Zou et al (2015) *Mol Biol Evol.* ;32, 132-43) and provide further support to these data. NMR or MD experiments are clearly beyond the present work, but these data should be discussed at the light of previous evidence.

Note:

All line numbers in the reviewer's response correspond to the new manuscript version. Manuscript changes are italicized, and indicated with 'tracked changes' when necessary for clarity. The abstract was shortened to comply with the 150 word limit and doi numbers are added for online-only journals.

Reviewers' comments:**Reviewer #1 (Remarks to the Author):**

n this manuscript by Pawlowski et al. the authors make use of ancestral sequence reconstruction combined with structural and biochemical studies to determine how enzymes that modify macrolides evolve towards a broader substrate range.

This is a well-written, clear and interesting manuscript that provides a strategy for evaluating the evolvability of resistance enzymes to new generations of antibiotic scaffolds that is of both practical (drug development) and more basic interest (evolution and mechanistic basis for enzyme substrate specificity).

We appreciate this very positive comment!

Comments:

1. Page 5, the use of the word of sub-functionalization I do not think is proper here. Sub-functionalization in the context of evolution of new functions means that one function is partitioned between different gene copies (i.e. an interallelic complementation).

We thank the reviewer for making this important distinction, and we have clarified our statement on lines 95-98.

Lines 95-98

Applying phylogenetics, structural biology, protein engineering, and in vitro enzyme assays to the Mph family, we evolve increased substrate range in MphI, which results from multiple non-obvious mutations that, in tandem, increase the catalytic rate towards cladinose containing macrolides.

2. Page 7: "The most parsimonious explanation is that MphI lost the ability to phosphorylate C3 cladinose macrolides."

Page 14: "Narrow range Mphs such as MphI lost the ability to modify C3 cladinose macrolides, likely because of lack of selection by diverse macrolide structures in the local environment accompanied by neutral drift."

Could another explanation not be that there was stronger selection for ability to provide resistance to specific macrolides and that this resulted in loss of activity towards some macrolides because of functional trade-offs?

We have considered this a possibility, but our results suggest that decreased specificity towards cladinose macrolides does not significantly increase resistance to natural descladinose macrolides (tylosin and spiramycin).

- 1) *mphI* does not confer increased resistance to descladinose macrolides compared to *mphB* (Supplementary Table 7).
- 2) The tylosin and spiramycin k_{cat}/K_m values of MphI are only 2-4x higher than for MphB. This difference does not provide increased resistance when expressed in *E. coli*.
- 3) The tylosin and spiramycin k_{cat}/K_m values of MphI mutants with expanded substrate range (Azi_1.1 and Ery_1) are only 2-6x higher than for MphI. All three genes confer the same level of spiramycin and tylosin resistance.

3. Page 7, 8, “specificity constant” I would not call it a specificity constant. Rather k_{cat}/K_m is an efficiency factor but the ratio of the k_{cat}/K_m for substrate 1 and k_{cat}/K_m for substrate 2 provides a specificity measure.

We have changed “specificity constant” to “catalytic constant” on lines 151-152.

Lines 151 – 152

The difference in catalytic constant (k_{cat}/K_m) values between C3 cladinose macrolides and descladinose macrolides is about 10^3 and $10^1 - 10^3$ respectively for MphI and MphK.

4. Page 8 and forward. Was the structural analysis based on the authors data or the published data or both?

We used our MphB and MphH crystal structures as well as previously published structures of MphA and MphB. We have clarified this on **lines 174 – 178, line 179, and 193 – 194.**

Lines 174 – 178

While preparing this manuscript, crystal structures of MphA and MphB were reported in complex with various macrolides, which we have included in our analysis of substrate specificity⁴⁴. We also attempted to crystalize MphI and MphK as apoenzymes, with several GTP analogues, and with spiramycin, tylosin, and telithromycin, but we were unsuccessful.

Line 179

As expected, the structures of MphB (from our study) and MphH showed that the enzymes adopt the bi-lobe kinase fold reminiscent of APH(2”) enzyme crystal structures...

Lines 193 – 194

We compared the overall structure of MphH with MphA (5IGI⁴⁴) and MphB (from ref⁴⁴; 5IGV) to identify possible conformational changes associated with macrolide binding.

5. Page 10-11. Could shorten considerably the first paragraph on page 10-11 since the result is negative (i.e. the authors state that: “Therefore, C3 cladinose macrolide specificity does not appear to be directed by nearby residues.”). Instead, the subsequent phylogenetic analysis provides an answer and one could go directly into that result.

We appreciate the author's suggestion for improving the readability and focus of our manuscript. We have removed the in depth analysis of residues (lines 220 – 229) and instead just referred to the figures that demonstrate this on line 219.

Lines 220 – 229 (removed)

For example, position 268 in MphH is an Ala, and this residue is positioned under the C3 cladinose ring (Fig. 4b, Supplementary Fig. 7b); the corresponding residue in MphI is Tyr280. However, a significant proportion of Mphs that efficiently phosphorylate C3 cladinose macrolides also have a Tyr or Phe at that position (e.g. MphB Tyr273), suggesting that this residue does not play a significant role in substrate specificity. Similarly, MphH contains Gly271 positioned against the C3 cladinose ring. The corresponding residue in MphI is Leu283; however, other Mphs with wide substrate specificity have a larger residue such as a Thr or Ser at that position (e.g. MphA Thr276, MphB Ser276). Finally, the C3 cladinose ring also contacts MphH residue Phe229, but this residue is not conserved among wide-specificity Mphs (e.g. MphA Ala234) and is equivalent to Leu241 in MphI.

6. Page 13. What is the structural/functional interpretation of the fact that the mutations mainly affected kcat rather than Km?

We have now included this interpretation on lines 312 - 320.

Lines 311 - 319

Amino acid residues positioned near, but not in, substrate binding sites can impact function by modulating protein dynamics^{55,56}. For example, combinations of mutations located outside the β -lactam binding pocket of β -lactamases expand substrate range by increasing conformational flexibility^{49,55-57}. Introduction of each mutation individually does not expand the substrate range of β -lactamases⁵⁶. In our study, M40A and P211L are located outside the MphI macrolide binding pocket and both mutations are required to confer resistance to C3 cladinose macrolides. Altogether, the binary M40A and P211L mutations may increase the conformation flexibility of the macrolide binding pocket, increasing the phosphorylation rate of C3 cladinose macrolides.

7. Page 6, change reveal to reveals

Thank you. We have made this suggestion on line 120.

Line 120

Analysis of the taxonomic distribution of each Mph homolog reveals that MphA, MphB, and MphE are widespread in Gram-negative bacteria...

8. Page 8 "...we attempted determined...." Should be either attempted or determined.

Thank you. We have made this suggestion on line 172.

Line 172

To better understand the molecular specificity of Mph enzymes against macrolides, we determined the 3-D structure of MphB as the apoenzyme, and three forms of MphH:

Reviewer #2 (Remarks to the Author):

Resistance to macrolide antibiotics is growing, threatening the usefulness of these essential antibacterial agents. Macrolide kinases (Mph) inactivate macrolides by phosphorylating an OH of the sugar moiety, preventing binding to the ribosome. Mph enzymes from non pathogenic bacteria are functionally diverse, being narrow range, in contrast to Mph's from pathogens. In this work, Pawlowski et al. report the molecular basis for expanded substrate specificity in MphI.

*A bioinformatic analysis of the distribution of Mphs reveals that Gram negative pathogens have acquired Mph genes through mobile elements. MphI is the only known Mph unable to confer C3-cladinose resistance. The authors study a series of Mph enzymes from *B. subtilis* and interrogate their substrate profile. MphI and MphK show poor (but not null) in vitro specificities to phosphorylate cladinose macrolides, that results in a small impact on the resistance phenotype. In contrast, MphB is broad spectrum and shows high catalytic efficiencies in modifying several macrolides.*

The crystal structure of unbound MphB and ligated to azithromycin and GDP does not reveals clues about the specificity in active site residues. Ancestral sequence reconstruction provided clues to identify 4 positions that were randomized in MphI, resulting in the selection of mutants showing a clear gain of function in vitro. This has led to the identification of possible two mutational hotspots that may drive the expansion of substrate spectrum of Mphs.

This is an excellent piece of work, combining different techniques. There are, however, a few aspects that are unclear at this point:

1. The authors do not report the crystal structure of MphI, that would have been the ideal complement to the herein reported structures of MphB and MphH. Have they been unable to obtain crystals of MphI?

Indeed, we did attempt to crystallize MphI as the apo-enzyme, with different GTP analogs, and with macrolide substrates (telithromycin, spiramycin, and tylosin). We were ultimately unable to attain this and had to move forward with the MphH-azithromycin co-structure. While this manuscript was in preparation, the structures of MphA and MphB co-crystallized with macrolides were published (PMID: 28416110). We used these structures to supplement our analysis. We have now indicated this on lines 174 – 178.

Lines 174 – 178

While preparing this manuscript, crystal structures of MphA and MphB were reported in complex with various macrolides, which we have included in our analysis of substrate specificity⁴⁴. We also attempted to crystallize MphI and MphK as apoenzymes, with several GTP analogues, and with spiramycin, tylosin, and telithromycin, but we were unsuccessful.

Homology modelling may be insufficient.

Our analysis of substrate specificity did not rely on homology modelling of MphI using MphA, MphB, or MphH. Instead, our analysis of substrate specificity relied on the two extensive multiple sequence alignments that were used to generate phylogenetic trees in Fig. 2 and Fig. 5a.

2. *The mutated residues are nearby, but not within the active site. Based on the crystal structures, the authors indirectly relate the impact of the substrate spectrum to conformational changes linked to substrate binding. Broadening of the substrate spectrum by second sphere mutations have been addressed by NMR (González et al (2016) Mol Biol Evol.;33, 1768-76) and Molecular Dynamics in beta-lactamases (Zou et al (2015) Mol Biol Evol. ;32, 132-43) and provide further support to these data. NMR or MD experiments are clearly beyond the present work, but these data should be discussed at the light of previous evidence.*

We appreciate the reviewer bringing these studies to our attention, and we have commented on the impact of conformational flexibility on substrate range on lines 311 - 319.

Lines 311 – 319

Amino acid residues positioned near, but not in, substrate binding sites can impact function by modulating protein dynamics^{55,56}. For example, combinations of mutations located outside the β -lactam binding pocket of β -lactamases expand substrate range by increasing conformational flexibility^{49,55-57}. Introduction of each mutation individually does not expand the substrate range of β -lactamases⁵⁶. In our study, M40A and P211L are located outside the MphI macrolide binding pocket and both mutations are required to confer resistance to C3 cladinose macrolides. Altogether, the binary M40A and P211L mutations may increase the conformation flexibility of the macrolide binding pocket, increasing the phosphorylation rate of C3 cladinose macrolides.